# Peripheral Vascular Disease and Carotid Artery Disease Are Associated with Decreased Bile Acid Excretion

**DOI:** 10.3390/bioengineering10080935

**Published:** 2023-08-07

**Authors:** Lior Charach, Gideon Charach, Eli Karniel, Leonid Galin, Dorin Bar Ziv, Lior Grossman, Irit Kaye, Itamar Grosskopf

**Affiliations:** 1Department of Internal Medicine B, Meir Medical Center, Kfar Saba 4428164, Israel; drliocharach@gmail.com (L.C.); eli.karniel.2017@gmail.com (E.K.); leonid.galin@gmail.com (L.G.); dorinbarziv@gmail.com (D.B.Z.); grossman.lior@gmail.com (L.G.); iritgutermann@gmail.com (I.K.); itamar.grosskopf@gmail.com (I.G.); 2Sackler School of Medicine, Tel Aviv University, Tel Aviv 6139001, Israel

**Keywords:** atherosclerosis, bile acids, carotid artery stenosis, low bile acid excretion, peripheral vascular disease

## Abstract

Low bile acid excretion (BAE) is associated with a higher risk of coronary artery disease (CAD) and cerebrovascular disease (stroke). This study investigated BAE in patients with peripheral vascular disease (PVD) and carotid artery disease (CA) and those without these diseases, compared to patients with CAD, stroke, or no evidence of atherosclerosis. Patients with complaints of chest pain-suspected CAD, syncope, stroke/TIA, severe headache, intermittent claudication, or falls were enrolled. All received a 4-day standard diet with 490 mg of cholesterol and internal standard copper thiocyanate. Fecal BAE was measured using gas–liquid chromatography. One hundred and three patients, sixty-eight (66%) men and thirty-five women (34%), mean age range 60.9 ± 8.9 years, were enrolled in this prospective, 22-year follow-up study. Regression analysis showed that advanced age, total BAE, and excretion of the main fractions were the only significant independent factors that predicted prolonged survival (*p* < 0.001). Twenty-two years’ follow-up revealed only 15% of those with BAE <262.4 mg/24 h survived, compared to >60% of participants without atherosclerosis and a mean BAE of 676 mg/24 h. BAE was lower in patients with polyvascular atherosclerosis than in those with involvement of 1–3 vascular beds. Pearson correlations were found between total BAE and various fractions of BA, as well as HDL cholesterol. BAE and short-term survival were decreased among patients with PVD compared to those with CAD or stroke. Low BAE should be considered a valuable and independent risk factor for PVD.

## 1. Introduction

Atherosclerosis is a well-known inflammatory process associated with cholesterol deposition within the arterial wall. Increased cholesterol levels are associated with a greater risk of atherosclerotic vascular disease. Pharmacological treatments to reduce cholesterol levels known to have favorable effects on morbidity and mortality include HMG-CoA reductase inhibitors and PCSK9 inhibitors [1,2]. Rodents did not develop experimental atherosclerosis despite a diet rich in cholesterol due to excreting large amounts of bile acids, as described in animal studies [2,3,4,5,6,7,8,9,10,11,12,13]. Similar outcomes were obtained in a study on New Zealand white rabbits and primates, which were also fed a cholesterol-rich diet [14]. In those studies, the animals with high excretion of cholesterol did not suffer from hypercholesterolemia, while the animals with low excretion had higher levels of cholesterol in the plasma [14]. Moreover, an inverse correlation was found between the level of hypercholesterolemia and the rate of bile acid excretion. These studies suggest that the atherogenic effect of a cholesterol-rich diet closely depends on the animal’s ability to excrete cholesterol as bile acids [12,13]. Thus, it is reasonable to hypothesize that reduced conversion of cholesterol to bile acids would induce cholesterol overload and probably accelerate the rate of atherosclerosis [6,7,8,9,10].

Bile acids are mostly composed of two primary (cholic acid and chendexoycholic acid) and two secondary bile acids (deoxycholic acid and lithocholic acid, cholic acid, and chendexoycholic acid), respectively [3,4,5,6,7,8]. Plasma triglycerides and HDL-cholesterol serve as precursors for bile acid biosynthesis in the liver. 

Bile acid synthesis and excretion are physiological mechanisms for the elimination of cholesterol from the body. Approximately 50% of the 800 mg of cholesterol synthesized daily by the human body is used to produce bile acids (BA) [3,4,5,6,7,8,9,10,11]. Synthesis of bile acids is controlled by a complex multistep process involving BA flux into the portal circulation, gall bladder function, conjugation to glycine or taurine, and regulation of elements such as the nuclear farnesoid X receptor, fibroblast growth factor-19, and Cyp7α1 [3,4,5,6,7,8,9,10,11,12,13]. One of these mechanisms is the transformation of cholesterol to bile acids via 7-α hydroxylase [2,7,9,10,12,13,14,15]. This enzyme has been shown to mediate the excretion of cholesterol from the plasma and intracellular space as well as eliminate bile acids.

Clinical research in healthy females has shown that HDL cholesterol and biliary saturation with cholesterol have a negative relationship [5,10]. Other research has shown that patients without non-CAD have a correlation between bile acid excretion and plasma triglycerides, which is in contrast to total cholesterol, LDL-c, and HDL-c [2,4,5,9,10,14,15]. This finding can be explained by the fast and improved intestinal absorption of triglycerides due to a surplus of bile acids, which are important factors for the emulsification of fats. In comparison, patients with CAD did not exhibit similar findings, probably due to a significantly lower amount of excreted bile acids [3]. The increased elimination of bile acids in the group of non-CAD patients can possibly be explained mainly by the higher activity and concentration of 7-α-hydroxylase [2,7,9,10,12,13,14,15], which has an important role in the metabolism of cholesterol. In contrast to it, CAD patients are unable to effectively elevate the concentration and activity of 7-α-hydroxylase [9,10,11]. 

The results of the previous [9,10] studies after a long follow-up showed up to a six-fold higher incidence of ischemic stroke among the CAD patients compared to the group of non-CAD patients (19% of patients versus 2.7%) and threefold higher mortality in the CAD group (25% versus 8% of patients) [9]. The hazard ratio of all-cause mortality in patients with BAE above 622 mg/24 h was 0.07 (*p* < 0.001) [10].

Patients with CAD eliminated significantly lower amounts of total BA, deoxycholic acid, and lithocholic acid compared to patients without CAD [11]. Low daily BAE (<415 mg) was associated with an increased risk of CAD mortality. We concluded that low BAE may be considered a valuable and independent risk factor for atherosclerosis. This was also found among patients with CAD and stroke. 

However, to date, there is no information about the association between BAE with PVD or CA stenosis. 

Atherosclerosis in two or more arterial beds is termed PVD [14,16]. PVD is a ubiquitous condition. It has received increased consideration in recent years due to growing clinical and research endeavors to include non-coronary atherosclerosis, in particular peripheral artery disease, carotid disease, and cerebrovascular disease. The relevance of PVD centers on its heightened risk for cardiovascular death, including myocardial infarction (MI), and ischemic stroke, a composite end-point known as major adverse cardiovascular events (MACE) [1,16,17,18]. The present study highlights the inherent risks associated with PVD by assessing the association between PVD of the legs and carotid arteries with reduced BAE.

Low-density lipoprotein cholesterol (LDL-c) can be decreased by enhancing fecal bile acid waste and by compensatory hepatic upregulation of bile acid synthesis [2,3,4,7,8,9,15,16,17,18].

## 2. Materials and Methods

One hundred and thirty-six patients were evaluated for the study, of whom thirty-three did not meet the inclusion criteria or did not consent to participate. One hundred and three patients admitted to the Internal Medicine Department of Tel Aviv Medical Center, Israel, were considered for enrollment in the study if they had chest pain suggestive of CAD or one of the following: syncope; stroke, including transient ischemic attack (TIA); severe headache; intermittent claudication; or falling. All prospective participants had ankle-brachial index measurements and arterial duplex examinations. Those with an ankle-brachial index <0.75 and/or >50% stenosis of the lower limb arteries and ≥50% stenosis of the carotid arteries were enrolled in the study. Exclusion criteria were age less than 40 years, malignancy, psychiatric disorder, liver disease, chronic renal failure, chronic obstructive lung disease, and acute infection, all of which can affect BAE.

The study included one hundred and three individuals. They were followed for up to 22 years, from January 1998 through December 2020. 

At the start of the study, each patient had a thorough physical examination. Medical history and electronic medical records were reviewed. Special consideration was given to CAD and non-hemorrhagic cerebrovascular disease (stroke) and their pertinent risk factors (diabetes mellitus, hypertension, hyperlipidemia, and smoking). All participants underwent extensive laboratory testing, including a complete blood count, glucose level, renal function (creatinine and urea), liver enzymes, creatinine kinase, troponin, and lipid profile. Following ECG, each patient had stress echocardiography or thallium scan tests according to clinical criteria and coronary angiography. Stool was tested for bile acid content, as previously described [4,5,6,9,10]. 

### 2.1. Endpoints and Assessments

The study endpoints in this study included overall survival in patients with PVD, isolated carotid artery stenosis, or both. Survival was determined as the interval from the first admission to the hospital to death from any cause or to the end of the study.

### 2.2. Study Protocol Based on Gilat and Grundy Methods

The study protocol was previously described [19,20]. Briefly, eligible participants provided 24 h stool collection. Copper isothiocyanate was used as the internal standard for further BA calculations. Each subject received 12 capsules of 75 mg copper isothiocyanate to be taken one per meal for 4 days prior to stool collection. Seven days before taking copper isothiocyanate, patients were put on a standard hospital diet containing 490 mg/day of cholesterol. Patients who did not follow the strict, prescribed diet were excluded from the study. 

### 2.3. Quantitative Determination of the Bile Acids in Feces

An adjustment of the method described by Grundy et al. and Gilat and Ronen [19,20] was used to quantitate bile acids in the stool. This method enables the complete recovery and separation of bile acids. The amount of bile acids was determined using 225 mg of copper isothiocyanate as an internal standard [19]. The method for determining free bile acids and their fractions was described in detail in previous studies [10,12,15].

Seven days prior to ingestion of the capsules, participants were placed on a diet containing approximately 500 mg of cholesterol daily. Quantitative measurements of fecal BAs were collected after separation from neutral steroids and saponification of taurine and glycine conjugates. Free BAs were extracted with chloroform–methanol. Thin-layer chromatography (TLC) was used in order to remove the extract from fatty acids and pigments. Following TLC, 5ά-cholestan was added as an internal standard for gas–liquid chromatography (GLC). We used the gas–liquid chromatography (GLC) method for the detection of bile acids, which was widely used more than 20 years ago (today liquid chromatography-mass spectrometry is the largely used method for the determination of BA isomers). The BAs were transformed into trimethyl silylalanine (TMS) ethers, which were quantified by GLC. Standards bile acids and 5ά-cholestane were obtained from Sigma Chemical Co. (St. Louis, MO, USA), pyridine super dried from E. Merck (Darmstadt, Germany), and hexamethyldisalazan and trimetylchlorosilane from Fluka AG (Buch SG, Switzerland). Thin-layer chromatography was carried out on 0.25 mm of Silica Gel layers (Merck). GLC of bile acid derivatives was performed by a gas–liquid chromatograph (Unigam, Cambridge, UK) equipped with a hydrogen flame ionization detector. A glass column 6 ft × 4 mm packed with 3% OV-1 on 100–120 mesh chromosorb W-HP (Alltech Associates, Inc., Deerfield, IL, USA) was used. Nitrogen with a flow rate between 30–60 mL/min was used as the carrier gas. The column temperature was 275 °C, and that of the flash heater and hydrogen flame ionization detector was 280 °C.

### 2.4. Statistical Methods

Data are described as numbers and percentages for nominal parameters and as mean ± standard deviation (SD) for continuous variables. The Shapiro–Wilk test was used to check the normality of the distribution of metric variables. Several variables were not normally distributed. For these, we used non-parametric tests, including Mann–Whitney and Kruskal–Wallis as appropriate, to compare between groups. Non-metric parameters were tested with the chi-square or Fisher’s exact test, as needed. Bonferroni correction was used to compare the two groups. Spearman’s correlation was calculated between total BAE and continuous variables. A Kaplan–Meier curve was used to evaluate the probability of an event at a given time interval. Cox regression was used to find the effect of several variables on the time of death. The receiver operating characteristic (ROC) curve showed the performance of a classification model at all classification thresholds. The best cut-off had the highest true positive rate together with the lowest false positive rate. All statistical analyses were performed using SPSS-27 (IBM Corp., Armonk, NY, USA).

## 3. Results

A total of one hundred and three individuals participated in this historical cohort and were followed for up to 22 years (see study flowchart below). Sixty-eight were males (66%) and thirty-five (34%) females, at a mean age range of 60.9 ± 8.9 years. In this cohort, 56 had CAD, 24 had stroke or TIA, and 23 had PVD of the legs with or without CA disease. A few patients had a combination of these, and 32 were free of vascular disease (Figure 1).

Table 1 presents the demographic, clinical, and relevant laboratory data. The mean BAE for the cohort was 484.9 ± 285.9 mg/24 h. The mean total cholesterol was 245.5 ± 54 mg/dL.

During the follow-up period, 64 patients survived and 39 died. BAE among survivors was significantly higher than among deceased patients (608.8 ± 290.9 mg/24 h vs. 281.5 ± 103.8 mg/24 h, respectively; *p* < 0.0001). The statistical cut-off showed that BAE >360 ± 4.83 mg/24 h was associated with a favorable prognosis.

To better understand the contribution of CA disease and PVD to the risk of death, the characteristics of three groups of patients were compared: Patients without PVD and without CA disease; patients with either PVD or CA disease; and patients with PVD and CA disease (Table 2). Total BAE was significantly higher in patients without PVD, CAD, CA disease, or ischemic stroke compared to patients with PVD (676.8 mg vs. 221.0 mg; *p* < 0.001). Total cholesterol, LDL cholesterol, and triglyceride levels, but not HDL cholesterol levels, did not differ significantly between groups (Table 2). The excretion of deoxycholic, lithocholic, and cholic acids was also significantly higher in patients without vascular disease (Table 2).

Figure 2 describes the combination of the various vascular bed atherosclerotic diseases and BAE per 24 h. Patients without atherosclerotic cardiovascular, cerebral, CA disease, or PVD of the lower extremities had the highest BAE (676.8 mg/24 h). Other columns show atherosclerotic process involvement at one to four polyvascular sites. The BAE of patients diagnosed with polyvascular atherosclerosis in four sites had the lowest BAE. Those groups with only one or more diseases were statistically significant from the others. This indicated that depending on diffuse atherosclerosis expression, BAE is lower when more vessel sites are involved (Table 2). Total BAE was more than 3.6 times higher among patients without vascular disease than among those with PVD, CAD, and ischemic stroke (676.8 ± 324.4 mg vs. 189.9 ± 30.3 mg).

BAE among patients with PVD or CAD and those without vascular disease did not differ according to major risk factors, such as type 2 diabetes, hypertension, kidney disease, hypercholesterolemia, and smoking. The groups did not differ in LVEF or the percentage of patients who used statins, antihypertensive medicines, aspirin, or clopidogrel.

Table 3 shows Spearman’s correlations between total BAE and various clinical, demographic, and laboratory parameters. Age at the beginning of the study had a negative correlation with total BAE (r = −0.202, *p* = 0.041) and LVEF (r = 0.239, *p* = 0.015). BA main fractions (cholic acid, deoxycholic acid, lithocholic, and HDL) were also positively correlated with total BAE (all *p* < 0.0001). No correlation was found between BAE or total cholesterol, LDL cholesterol, triglycerides, BMI, and such risk factors for vascular disease, such as diabetes, hypertension, or smoking.

Figure 3 and Figure 4 show Kaplan–Meier cumulative survival curves between groups with and without PVD, the survival of patients with PVD of the legs, PVD of the carotid artery, and patients without PVD.

Survival after 22 years among patients with PVD and carotid artery stenosis was 15% (Figure 3), indicating mortality was 4.3-fold higher compared to those without evidence of PVD and CA disease (including carotid artery stenosis), without stroke or CAD (*p* = 0.01). Survival of patients with carotid artery stenosis alone was 22% (Figure 4), indicating mortality was 2.7-fold higher compared to the groups mentioned above. Seventy-one patients had either PVD, CA, CVA, or CAD. Among these patients, 60.4% died; 3.8% died due to infections and 3.8% because of accidents (falls, motor vehicle accidents); 59.8% died because of CA, PVD, CAD, or CVA single or polyvascular disease.

Multivariate Cox regression analysis was used to assess the association between survival and possible predictor variables. Parameters that were statistically significant in the univariate analysis were entered into a multivariate Cox regression (Table 4). After adjustment for age, LVEF, stroke, and total BAE had significant hazard ratios and each independently influenced survival (*p* = 0.037, *p* = 0.022, and *p* = 0.008, respectively). These characteristics indicate a strong relationship between survival and disease-related mortality.

## 4. Discussion

This study examined fecal BAE in patients with PVD of the legs and carotid arteries compared to patients without PVD. The results show an association between low BAE and the severity of PVD as well as CA disease in terms of lower survival rates. In previous research, we showed that BAE is positively correlated with CAD and with the occurrence of stroke [9,10]. Remarkably, the correlation of BAE with PVD is stronger than the correlation with stroke and CAD. This suggests that the more diffuse the atherosclerotic process involving the peripheral vasculature, the lower the BAE. This new finding was seen when the excretion of the main BA fractions was measured. Excretion of deoxycholic, lithocholic, and cholic acids was significantly higher in individuals without evidence of peripheral atherosclerosis. Importantly, patients without PVD or CA disease excreted 60% more total BA, deoxycholic acid, and lithocholic acid than patients who had only PVD or CA disease, and those excreted 20% to 60% more BA (total and fractions) than patients with PVD and CA disease. Patients without PVD and without CA disease had a total BAE >566 mg/24 h in comparison to patients with only PVD or PVD plus CA disease, who had a BAE <296 mg/24 h. These results may suggest that high levels of BAE may be protective against PVD.

The findings of this study show that a reduced level of fecal BAE is strongly and independently associated with mortality in PVD, including CA disease. Similar findings were reported in patients with CAD and stroke [10,11,12,13,17,18]. The predictive value of BAE in PVD and CA stenosis or occlusion was stronger than for CAD and stroke.

This study adds additional evidence emphasizing the important role of fecal BAE among patients with vascular disease (PVD, CA stenosis, CAD, or stroke.

The question that remains is whether a lower BAE can be used as a quantitative indicator of the dissemination and severity of the atherosclerotic process, such as the Gensini score [8] in CAD. This has yet to be determined.

No correlations were found between BAE and total cholesterol, LDL cholesterol, triglycerides, BMI, or other risk factors for vascular disease, such as diabetes, hypertension, or smoking. Additional studies are needed in order to validate our contention not only in feces but also in blood.

Although existing drugs, such as statins and PCSK9, have proven their efficacy, cardiovascular diseases continue to be the cornerstone of morbidity and mortality in industrialized countries. It is significant to indicate that the ability to eliminate large amounts of bile acids impedes the development of stroke and ischemic heart disease [8,9,12,13,14,17], and as proven in the current study, carotid arteries and PVD as well.

Looking into the future: Bile acid measurements in the blood are a much simpler technique and require less patient compliance. (It is not necessary to collect the stools for 3 days). It was reported that low concentrations of BA in the blood were prognostic markers of CAD in people [8]. It was found that the amount of bile acids in the serum was a prognostic factor for atheroma in coronary arteries proven on angiograms. A substantial number of patients develop atherosclerosis despite statin therapy and reduced cholesterol levels. We suggest that, by reducing cholesterol synthesis and enhancing the utility of cholesterol, an additive anti-atherosclerosis effect might be obtained. By combining statins or PCSK9 inhibitors with bile acid sequestrants treatment affecting bile acid and cholesterol absorption, optimal management of dyslipidemia may potentially be achieved. A cholesterol-lowering effect can be gained by reducing cholesterol synthesis or by elevating bile acid excretion in the stool (ileal sodium-dependent bile acid transporter inhibitors).

This study was limited by a small sample. Validating the association of PVD with BAE requires larger prospective trials, which will help to elucidate whether the measurement of BAE can serve as a reliable indicator in the early stages of PVD and other manifestations of atherosclerosis in an asymptomatic general population. The current study did not include information on medical treatment during the follow-up period, which could affect the results. We consider this lack of data a significant limitation of the study.

Strengths of this study include showing the protective effect of BAE in the pathogenesis of PVD and CA disease. The long-term follow-up could suggest that a single assessment of BAE might indicate the future development of clinically significant PVD and CA disease at a very early stage.

## 5. Conclusions

The current study found strong relations between low excretion of total BAE and its main fractions (deoxycholic and lithocholic acids) and PVD and CA disease. Remarkably, as more sites are involved with atherosclerosis, BAE is lower. Diminished BAE, as cholesterol utilization, is an independent risk factor for PVD.

## Figures and Tables

**Figure 1 bioengineering-10-00935-f001:**
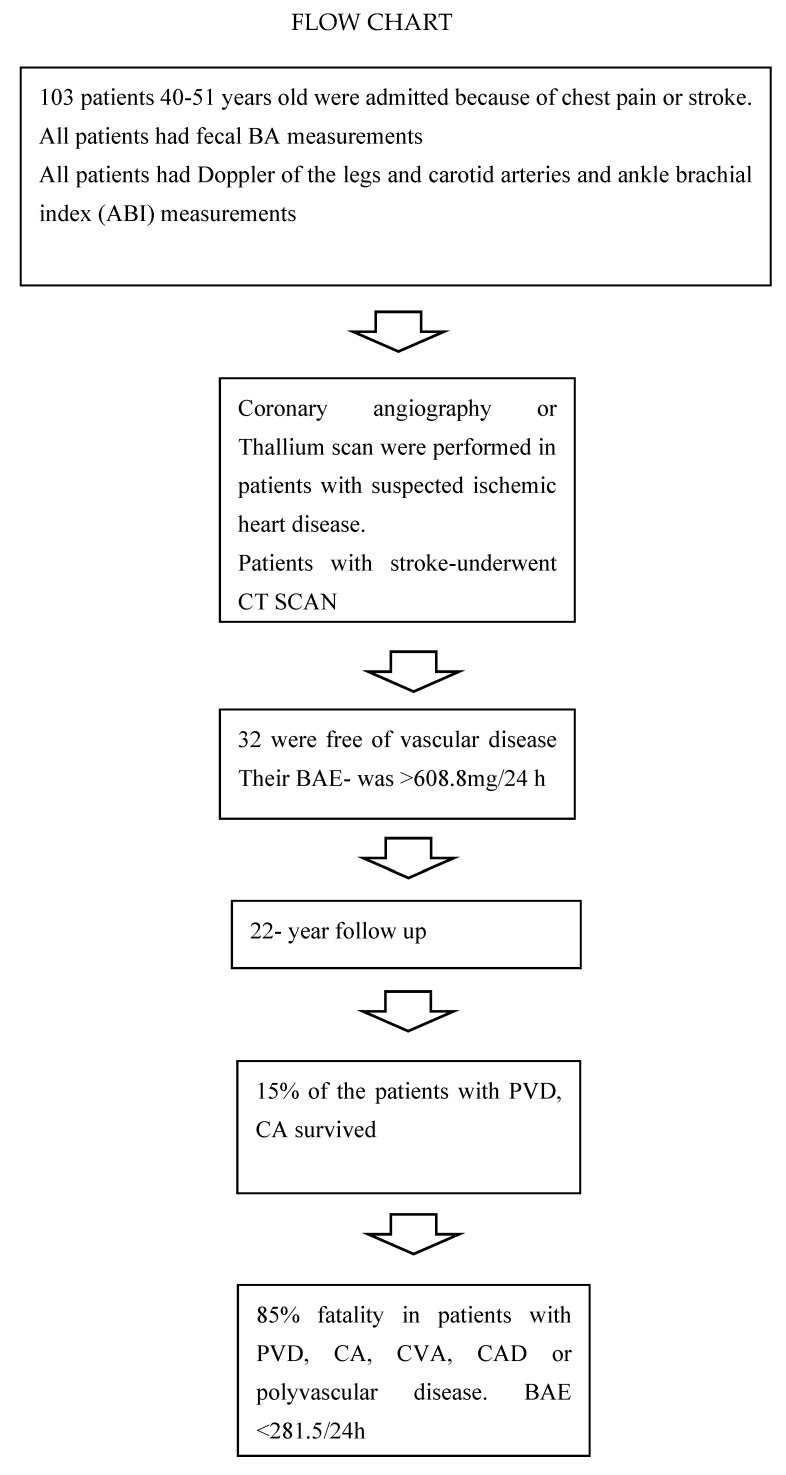
Venn diagram illustrating the comorbidities and survival (S) among the patient sample.

**Figure 2 bioengineering-10-00935-f002:**
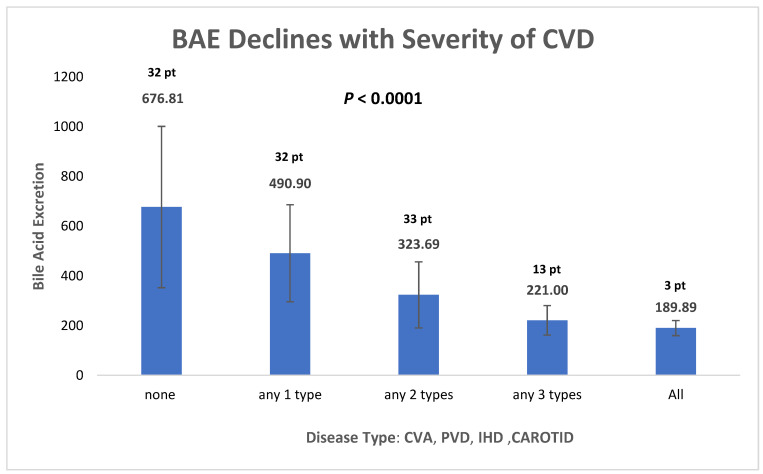
Increasing severity of CVD has a negative effect on BAE.

**Figure 3 bioengineering-10-00935-f003:**
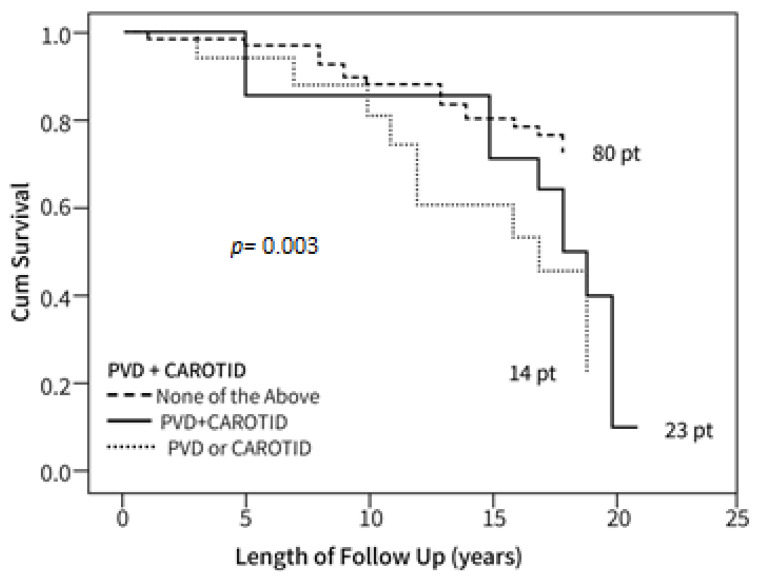
Kaplan–Meier survival curve of patients with or without a combination of PVD of the legs and carotid artery disease. Individual timepoints for patient follow-up are shown.

**Figure 4 bioengineering-10-00935-f004:**
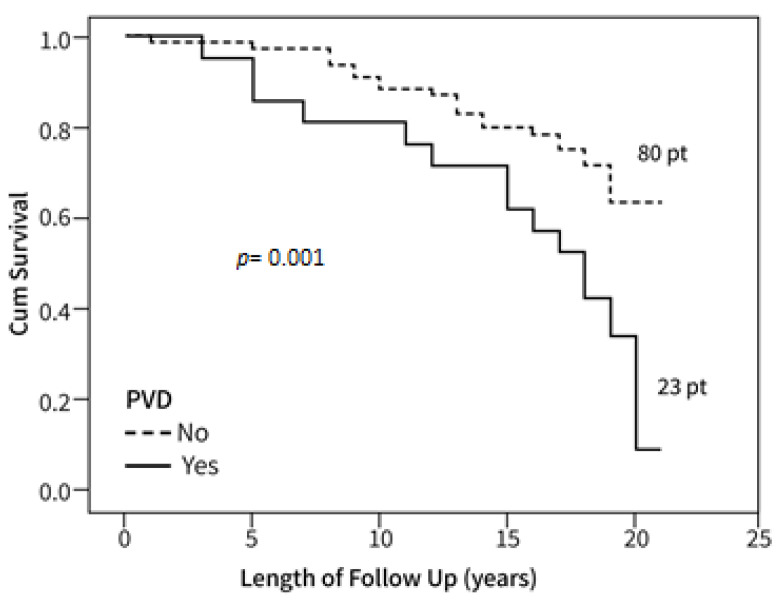
Kaplan–Meier survival curve of patients with or without PVD. Individual timepoints for patient follow-up are shown.

**Table 1 bioengineering-10-00935-t001:** Patient demographic, clinical, and laboratory data.

Characteristics	N = 103	% or ±SD
Sex, male	69	67.0%
Age at the beginning of the study (years)	60.9	±8.9
Smoking	31	30.1%
Diabetes mellitus	34	33.0%
Cerebrovascular accident	24	23.3%
Ischemic heart disease	56	54.4%
Hypertension	42	40.8%
Body mass index	27.2	±4.3
Total bile acid excretion (mg/24 h)	484.9	±285.9
Cholic acid (mg/24 h)	45.8	±93.5
Chenodeoxycholic (mg/24 h)	12.8	±16.0
Deoxycholic acid (mg/24 h)	279.3	±192.3
Lithocholic acid (mg/24 h)	155.0	±104.4
Left ventricular ejection fraction	56.1	±6.9
Triglycerides (mg/dL)	219.8	±92.1
Low-density cholesterol (mg/dL)	151.0	±45.0
High-density cholesterol (mg/dL)	47.8	±12.5
Total cholesterol (mg/dL)	245.5	±54.4

**Table 2 bioengineering-10-00935-t002:** Characteristics of patients without PVD and without carotid disease, patients with either PVD or carotid disease, and patients with PVD and CA disease.

Variable	No PVD + No CA Disease		PVD or CA Disease		PVD + CA Disease		*p*-Value **	Bonferroni ^a^
	N = 72	%	N = 17	%	N = 14	%		
Ischemic heart disease	34	47.2%	11	64.7%	11	78.6%	0.063	
Sex	50	69.4%	9	52.9%	10	71.4%	0.339	
Smoking	25	34.7%	4	23.5%	2	14.3%	0.254	
Hypertension	25	34.7%	10	58.8%	7	50.0%	0.144	
Diabetes mellitus	22	30.6%	8	47.1%	4	28.6%	0.399	
CVA	6	8.3%	7	41.2%	11	78.6%	<0.0001	0 ≠ 1 ≠ 2
	Median	Range	Median	Range	Median	Range		
Age, years	61.0	42–78	66	41–78	60.5	47–76	0.256	
LVEF (%)	60.0	35–70	55	38–60	57.5	45–65	0.118	
BMI (kg/m^2^)	27.0	19–40	28	23–35	26.0	23–35	0.412	
Cholic acid (mg/24 h)	18.8	0–445	9	0–154	4.7	0–156	0.020	0 ≠ 1, 2
Chenodeoxch acid (mg/24 h)	8.2	0–68	6	0–36	6.5	0–39	0.465	
Deoxych acid (mg/24 h)	253.2	114–923	160	28–496	109.5	53–270	<0.0001	0 ≠ 1,2
Lithocholic acid (mg/24 h)	156.5	6.5–762	91	16–241	76.0	9–141	<0.0001	0 ≠ 1
Age (y) at end of follow-up	78.0	58–88	78	59–88	77.0	66–90	0.779	
Triglycerides (mg/dL)	214.0	20–468	218	127–404	215.5	140–530	0.803	
LDL (mg/dL)	133.5	66–316	152	108–190	152.0	62–256	0.691	
HDL (mg/dL)	48.5	31–75	45	32–68	36.0	20–51	0.001	0 ≠ 1
Total Chol (mg/dL)	233.0	121–396	245	178–292	244.0	113–341	0.586	
Total BA (mg/24 h)	485.5	213–1698	290	170–734	180.5	166–425	<0.0001	0 ≠ 1,2

^a^ Bonferroni, 0 ≠ 1 ≠ 2 = all vs. all; 0 ≠ 1,2 = No; PVD + No CA disease vs. others; 0 ≠ 1 = no PVD + no CA disease vs. PVD + CA disease artery disease; CVA = cerebrovascular accident; LVEF = left ventricular ejection fraction. ** Chi-square for non-metric variables or Kruskal–Wallis for continuous variables.

**Table 3 bioengineering-10-00935-t003:** Spearman’s correlations between selected continuous parameters and total BAE (N = 103).

Parameter	R	*p*-Value
Age at study onset	−0.202	0.041
Age at study end	−0.182	0.066
Body mass index	0.033	0.738
Left ventricular ejection fraction	0.239	0.015
Cholic acid	0.405	0.000
Chenodeoxycholic acid	0.165	0.095
Deoxycholic acid	0.910	0.000
Lithocholic acid	0.770	0.000
Triglycerides	0.034	0.736
Low-density lipoproteins	−0.167	0.093
High-density lipoproteins	0.358	0.000
Total cholesterol	−0.093	0.351

**Table 4 bioengineering-10-00935-t004:** Multivariate survival analysis to assess survival predictors.

Characteristic	Sig.	Hazard Ratio	95% CI	
			Lower	Upper
Age at study onset	<0.001	1.208	1.119	1.305
Sex, male	0.435	1.437	0.912	3.636
Left ventricular ejection fraction	0.055	0.955	0.909	1.017
Deoxycholic acid	0.072	1.012	0.999	1.025
Lithocholic acid	0.438	1.005	0.992	1.016
Total bile acid excretion	0.007	0.983	0.971	0.995
Ischemic heart disease	0.727	0.848	0.337	2.133
Cerebrovascular accident	0.027	2.909	0.126	7.515
Peripheral vascular disease	0.476	1.474	0.507	4.282
Carotid artery disease	0.475	0.688	0.247	1.919

## Data Availability

Upon request, Meir Medical Center, Email: drliorcharach@gmail.com.

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
