# Peer review of "Peripheral Vascular Disease and Carotid Artery Disease Are Associated with Decreased Bile Acid Excretion"

_bioengineering, 2023, doi:10.3390/bioengineering10080935_

Round 1

Reviewer 1 Report

This study was well conducted, and was also well written. 

Author Response

  • Not all abbreviations have been introduced in the main text. These should be replaced.

       Abbreviations are now introduced in the main text

  • What were the reasons for excluding people under 40 from the study?

       Atherosclerosis process (CAD or stroke) is very uncommon in people under the 40 y ans usually a consequence of complex risk factors.

  • A flow chart for the sample selection process would be useful.

       Flow chart was included in the results section

  • Does the use of "death from any cause" as an outcome variable bias the results? Did death occur independently of cardiovascular disease or as a result of an accident?

      The cause of death in each and every case was not identified, however, in the years 2000-2020 accidents were the 7th most common cause of death in the age group  45-64 years old and the 10th most common  cause of death in the age group  65-74 years old

      71 patients had PVD, CA, CVA, CAD.. 60.4% died. 3.8% died because of infections and 3.8 because of accidents (fall,motor vehicle accidents)

  • Lines 119-120 belong more to the section on materials and methods.

       This was moved – first sentence in material and methods section

  • Figure 1 is difficult to read because of the small font size.

       The font was enlarged

  • In Figure 1, in addition to the characteristics of the whole population, it would be useful to show a comparison of the characteristics of survivors and non-survivors. In addition, the presentation of % and quantitative data should be separated.

      In Figure 1 we added survival rates in percent's (s-%)

  • What is the definition of smoker, hypertensive, diabetic, hypercholesterolemia in this study?

       Smokers were current smokers or in the past 10 years.

       Diabetics were patients who received antidiabetic treatment and their fasting glucose levels were above 125mg in two different measurements or Occasional plasma glucose value of ≥ 200 mg/dl

      Hypercholesterolemia was defined according the AHA guidelines as well as recieving  treatment with a statin and or ezetimibe

  • For Table 2, the units of measurement are missing, as well as whether sex refers to men or women.

        Units were added in the table 2

  • In the case of Figure 2, a p for trend analysis might be useful, as well as the indication of sample numbers for the subgroups.

 P for trend analysis was p<0.0001 for all subgroups

Group without any of the diseases CAD, CVA, PVD and CA was statistically significant from the others.

Those with only one or more diseases  were statistically significant from the others

 The sample numbers of subgroups were added in the curves

  • For Figures 3 and 4, what sample sizes were used for the analysis? It is recommended that these are indicated.

       The group samples were added

  • What is the p-value associated with a 4.3-fold risk on line 180 and a 2.7-fold risk on line 183?

       We added p=0.01-Results section under the Fig 4

  • Where are the results of the univariate analysis referred to in line 186 shown?

Here is enclosed univariate analysis (table). We think that this table will not add for and erstanding because all variables were included in multivariable analysis…

Univariate Variables in the Equation

B

SE

Wald

df

Sig.

HR

95.0% CI for HR

Lower

Upper

LVEF

-0.068

0.021

10.567

1

0.001

0.935

0.897

0.974

DeoxchA

-0.007

0.002

16.337

1

0.000

0.993

0.990

0.997

Litho

-0.013

0.003

23.774

1

0.000

0.987

0.982

0.992

TotBA

-0.006

0.001

23.748

1

0.000

0.994

0.992

0.997

IHD

1.085

0.371

8.564

1

0.003

2.960

1.431

6.123

CVA

0.855

0.326

6.883

1

0.009

2.352

1.242

4.456

PVD

1.037

0.326

10.114

1

0.001

2.821

1.489

5.344

CAROT

0.704

0.329

4.566

1

0.033

2.021

1.060

3.853

ages

0.186

0.030

37.995

1

0.000

1.204

1.135

1.278

gender

-0.668

0.327

4.168

1

0.041

0.513

0.270

0.974

These variables were adjusted to age and gender in multivariable analysis

  •  

  What explains the risk-reducing effect of "Left ventricular ejection fraction" and "Cerebrovascular accident" (HR<1) shown in Table 4? Furthermore, what was the outcome (binary) variable for these analyses? The title does not adequately inform the reader. I suggest revising it.

      Recalculation was performed by senior statistician- Left ventricular ejection fraction was not significant p=0.055. CVA   HR was 2.909. The table 4 was modified

The title was changed -Table 4:  Multivariate Survival analysis (cox regression) to assess survival predictors

Submission Date

21 May 2023

Date of this review

14 Jun 2023 12:00:19

Reviewer 2 Report

I read with interest the manuscript written by Charach et al., regarding the inherent risks associated with PVD by assessing the association of PVD of the legs and carotid arteries with reduced BAE. The author found a strong relationship between low excretion of total BAE and its main fractions (deoxycholic and lithocholic acids) and PVD and CA disease.

The manuscript is well written, very easy to read, with an innovative idea and a very long-term follow up.

However, I have some suggestions that can improve the quality of the manuscript:

1. The authors should mention in the Materials and Methods Section, how the patients were monitored, at what interval they had to return for control. Moreover, it is very important if the degree of stenosis for PVD and CA disease was monitored during the 22 years.

2. It will be very interesting, if the authors have quantified at the control visits the stage of PVD and CA (degree of stenosis) and to introduce the new data in the statistical analysis, to verify the correlation between BAE and the evolution of stenosis in terms of PVD and THAT

3. I suggest improving the number of references, as well as the Discussion section.

Author Response

I read with interest the manuscript written by Charach et al., regarding the inherent risks associated with PVD by assessing the association of PVD of the legs and carotid arteries with reduced BAE. The author found a strong relationship between low excretion of total BAE and its main fractions (deoxycholic and lithocholic acids) and PVD and CA disease.

The manuscript is well written, very easy to read, with an innovative idea and a very long-term follow up.

However, I have some suggestions that can improve the quality of the manuscript:

  1. The authors should mention in the Materials and Methods Section, how the patients were monitored, at what interval they had to return for control. Moreover, it is very important if the degree of stenosis for PVD and CA disease was monitored during the 22 years.

We did not have access to clinical data. This work was aimed at studying survival rate

  1. It will be very interesting, if the authors have quantified at the control visits the stage of PVD and CA (degree of stenosis) and to introduce the new data in the statistical analysis, to verify the correlation between BAE and the evolution of stenosis in terms of PVD and THAT

This is an important remark, however, in this work only survival was studied because of limited data was available  during monitoring

  1. I suggest improving the number of references, as well as the Discussion section.

Discussion section was edited in order to improve numbering of references. These were typed in red

The manuscript was expanded to 4000 words

Reviewer 3 Report

Thank you for the opportunity to review a very interesting article entitled "Peripheral vascular disease and carotid artery disease are associated with decreased bile acid excretion".

The research results provide an opportunity to go a step further in diagnosing peripheral vascular disease and carotid artery disease .

Very well-written manuscript:

- reliable study, 

- clearly documented with tables and figures, 

- correctly selected statistical methods,

- well described statistical analysis.

Very well-written manuscript. I recommend correcting minor stylistic and substantive errors. Corrections are included in the attached document.

Author Response

Thank you for your for your important remarks

Reviewer 4 Report

Charach and colleagues investigated bile acid excretion (BAE) in patients with peripheral vascular disease (PVD) with or without carotid artery disease, compared with patients with CAD who had had a stroke or without signs of atherosclerosis. A total of 103 patients (66% male, mean age 60.9±8.9 years) were enrolled in a 22-year prospective follow-up study. The regression analyses used showed that advanced age, total BAE and major fraction excretion were the only significant independent factors predicting prolonged survival (p<0.001). Only 15% of participants with BAE<262.4 mg/24h survived compared to >60% of participants without atherosclerosis with a mean BAE of 676 mg/24h. Pearson correlation was found between total BAE and different fractions of BA and HDL cholesterol. BAE and short-term survival were reduced in patients with PVD compared to those with CAD or stroke. Low BAE was found to be an independent risk factor for PVD.

Comments and suggestions:

·         Not all abbreviations have been introduced in the main text. These should be replaced.

·         What were the reasons for excluding people under 40 from the study?

·         A flow chart for the sample selection process would be useful.

·         Does the use of "death from any cause" as an outcome variable bias the results? Did death occur independently of cardiovascular disease or as a result of an accident?

·         Lines 119-120 belong more to the section on materials and methods.

·         Figure 1 is difficult to read because of the small font size.

·         In Figure 1, in addition to the characteristics of the whole population, it would be useful to show a comparison of the characteristics of survivors and non-survivors. In addition, the presentation of % and quantitative data should be separated.

·         What is the definition of smoker, hypertensive, diabetic, hypercholesterolemia in this study?

·         For Table 2, the units of measurement are missing, as well as whether sex refers to men or women.

·         In the case of Figure 2, a p for trend analysis might be useful, as well as the indication of sample numbers for the subgroups.

·         For Figures 3 and 4, what sample sizes were used for the analysis? It is recommended that these are indicated.

·         What is the p-value associated with a 4.3-fold risk on line 180 and a 2.7-fold risk on line 183?

·         Where are the results of the univariate analysis referred to in line 186 shown?

·         What explains the risk-reducing effect of "Left ventricular ejection fraction" and "Cerebrovascular accident" (HR<1) shown in Table 4? Furthermore, what was the outcome (binary) variable for these analyses? The title does not adequately inform the reader. I suggest revising it.

Author Response

Not all abbreviations have been introduced in the main text. These should be replaced.

       Abbreviations are now introduced in the main text

  • What were the reasons for excluding people under 40 from the study?

       Atherosclerosis process (CAD or stroke) is very uncommon in people under the 40 y ans usually a consequence of complex risk factors.

  • A flow chart for the sample selection process would be useful.

       Flow chart was included in the results section

  • Does the use of "death from any cause" as an outcome variable bias the results? Did death occur independently of cardiovascular disease or as a result of an accident?

      The cause of death in each and every case was not identified, however, in the years 2000-2020 accidents were the 7th most common cause of death in the age group  45-64 years old and the 10th most common  cause of death in the age group  65-74 years old

      71 patients had PVD, CA, CVA, CAD.. 60.4% died. 3.8% died because of infections and 3.8 because of accidents (fall,motor vehicle accidents)

  • Lines 119-120 belong more to the section on materials and methods.

       This was moved – first sentence in material and methods section

  • Figure 1 is difficult to read because of the small font size.

       The font was enlarged

  • In Figure 1, in addition to the characteristics of the whole population, it would be useful to show a comparison of the characteristics of survivors and non-survivors. In addition, the presentation of % and quantitative data should be separated.

      In Figure 1 we added survival rates in percent's (s-%)

  • What is the definition of smoker, hypertensive, diabetic, hypercholesterolemia in this study?

       Smokers were current smokers or in the past 10 years.

       Diabetics were patients who received antidiabetic treatment and their fasting glucose levels were above 125mg in two different measurements or Occasional plasma glucose value of ≥ 200 mg/dl

      Hypercholesterolemia was defined according the AHA guidelines as well as recieving  treatment with a statin and or ezetimibe

  • For Table 2, the units of measurement are missing, as well as whether sex refers to men or women.

        Units were added in the table 2

  • In the case of Figure 2, a p for trend analysis might be useful, as well as the indication of sample numbers for the subgroups.

 P for trend analysis was p<0.0001 for all subgroups

Group without any of the diseases CAD, CVA, PVD and CA was statistically significant from the others.

Those with only one or more diseases  were statistically significant from the others

 The sample numbers of subgroups were added in the curves

  • For Figures 3 and 4, what sample sizes were used for the analysis? It is recommended that these are indicated.

       The group samples were added

  • What is the p-value associated with a 4.3-fold risk on line 180 and a 2.7-fold risk on line 183?

       We added p=0.01-Results section under the Fig 4

  • Where are the results of the univariate analysis referred to in line 186 shown?

Here is enclosed univariate analysis (table). We think that this table will not add for and erstanding because all variables were included in multivariable analysis…

Univariate Variables in the Equation

B

SE

Wald

df

Sig.

HR

95.0% CI for HR

Lower

Upper

LVEF

-0.068

0.021

10.567

1

0.001

0.935

0.897

0.974

DeoxchA

-0.007

0.002

16.337

1

0.000

0.993

0.990

0.997

Litho

-0.013

0.003

23.774

1

0.000

0.987

0.982

0.992

TotBA

-0.006

0.001

23.748

1

0.000

0.994

0.992

0.997

IHD

1.085

0.371

8.564

1

0.003

2.960

1.431

6.123

CVA

0.855

0.326

6.883

1

0.009

2.352

1.242

4.456

PVD

1.037

0.326

10.114

1

0.001

2.821

1.489

5.344

CAROT

0.704

0.329

4.566

1

0.033

2.021

1.060

3.853

ages

0.186

0.030

37.995

1

0.000

1.204

1.135

1.278

gender

-0.668

0.327

4.168

1

0.041

0.513

0.270

0.974

These variables were adjusted to age and gender in multivariable analysis

  •  

  What explains the risk-reducing effect of "Left ventricular ejection fraction" and "Cerebrovascular accident" (HR<1) shown in Table 4? Furthermore, what was the outcome (binary) variable for these analyses? The title does not adequately inform the reader. I suggest revising it.

      Recalculation was performed by senior statistician- Left ventricular ejection fraction was not significant p=0.055. CVA   HR was 2.909. The table 4 was modified

The title was changed -Table 4:  Multivariate Survival analysis (cox regression) to assess survival predictors

Round 2

Reviewer 2 Report

no further comments.

Reviewer 4 Report

I accept the Authors' answers to my questions/suggestions.